# Best Procedure for the Management of Common Bile Duct Stones via the Papilla: Literature Review and Analysis of Procedural Efficacy and Safety

**DOI:** 10.3390/jcm9123808

**Published:** 2020-11-25

**Authors:** Shigeto Ishii, Hiroyuki Isayama, Mako Ushio, Sho Takahashi, Wataru Yamagata, Yusuke Takasaki, Akinori Suzuki, Kazushige Ochiai, Ko Tomishima, Ryo Kanazawa, Hiroaki Saito, Toshio Fujisawa, Shuichiro Shiina

**Affiliations:** Department of Gastroenterology, Graduate School of Medicine, Juntendo University, 2-1-1 Hongo, Bunkyo-ku, Tokyo 113-8421, Japan; sishii@juntendo.ac.jp (S.I.); m-ushio@juntendo.ac.jp (M.U.); sho-takahashi@juntendo.ac.jp (S.T.); w.yamagata.mx@juntendo.ac.jp (W.Y.); ytakasa@juntendo.ac.jp (Y.T.); suzukia@juntendo.ac.jp (A.S.); k.ochiai.qd@juntendo.ac.jp (K.O.); tomishim@juntendo.ac.jp (K.T.); rkanaza@juntendo.ac.jp (R.K.); hiloaki@juntendo.ac.jp (H.S.); t-fujisawa@juntendo.ac.jp (T.F.); sshiina@juntendo.ac.jp (S.S.)

**Keywords:** endoscopic sphincterotomy, endoscopic papillary balloon dilation, common bile duct stones, bleeding, post-ERCP pancreatitis

## Abstract

Background: Endoscopic management of common bile duct stones (CBDS) is standard; however, various techniques are performed via the papilla, and the best procedure in terms of both efficacy and safety has not been determined. Methods: Endoscopic procedures were classified into five categories according to endoscopic sphincterotomy (EST) and balloon dilation (BD): (1) EST, (2) endoscopic papillary BD (≤10 mm) (EPBD), (3) EST followed by BD (≤10 mm) (ESBD), (4) endoscopic papillary large BD (≥12 mm) (EPLBD), and (5) EST followed by large BD (≥12 mm) (ESLBD). We performed a literature review of prospective and retrospective studies to compare efficacy and adverse events (AEs). Each procedure was associated with different efficacy and AE profiles. Results: In total, 19 prospective and seven retrospective studies with a total of 3930 patients were included in this study. For EST, the complete stone removal rate at the first session, rate of mechanical lithotripsy (ML), and rate of overall AEs in EST were superior to EPBD, but a higher rate of bleeding was found for EST. Based on one retrospective study, complete stone removal rate at the first session, rate of ML, and rate of overall AEs were superior for ESBD vs. EST, and the rate of bleeding for the former was also lower. Complete stone removal rate at the first session and rate of ML for ESLBD were superior to those for EST, with no significant difference in rate of AEs. For EST vs. EPLBD, complete stone removal rate at the first session and rate of ML were superior for the latter. For EPLBD vs. ESLBD, the efficacy and safety were similar. Conclusions: ESBD is considered the best procedure for the management of small CBDS, but strong evidence is lacking. For large CBDS, both ESLBD and EPLBD are similar.

## 1. Introduction

Endoscopic management of common bile duct stones (CBDS) is standard, and there are various techniques for opening the orifice of the papilla for their removal. We considered these techniques from the point of view of efficacy, safety, and ease of the procedure, to investigate which is best.

Since endoscopic sphincterotomy (EST) was first developed in 1973 by Kawai as endoscopic treatment for bile duct stones [1], it has been the standard procedure for managing CBDS. However, EST is associated with a risk of bleeding, perforation, and dysfunction of the papillary sphincter, and an appropriate incision requires a high level of skill. The removal of stones is sometimes difficult after EST, because of an inadequate opening. An adequate EST incision is required for easy stone retraction, but a larger incision increases the risk of bleeding and perforation. Endoscopic papillary (EP) balloon dilation (BD) was introduced as an alternative method to EST in 1984 by Staritz [2]. EPBD is a less complicated procedure and does not require a high level of skill to achieve sufficient papillary dilation. EPBD has an extremely low incidence of bleeding and perforation, and the function of the sphincter of Oddi is preserved [3]. However, EPBD has a high risk of post-endoscopic retrograde cholangiopancreatography pancreatitis (PEP) [4,5]. Although the incidence of PEP is low in specialized EPBD centers [6], clinical trials in multiple centers have revealed that EPBD is associated with a significantly higher incidence of PEP compared with EST [7]. Typically, EST and EPBD are performed for removing small CBDS, with some reports on the value of using EST and EPBD in combination. EST followed by BD is a relatively new method for the management of small bile duct stones. We have reported on the safety and efficacy of minimal EST followed by balloon dilation (ESBD), which decreases the incidence of hemorrhage and is easier to use than EST for the removal of small CBDS [8].

Large bile duct stones are more difficult to remove using conventional procedures, such as EST and EPBD. The length of the EST incision tends to be longer than for small CBDS, and the rates of bleeding and perforation are increased. Additional large BD (≥12 mm), called EP large BD (EPLBD), is effective for the removal of large stones and can be an alternative to conventional methods [9]. EPLBD can reduce the rate of mechanical lithotripsy (ML) and shortens procedure time without increasing the incidence of adverse events (AEs) [10,11]. There are two types of EPLBD, classified according to whether EST is performed: EST followed by large BD (ESLBD) and EPLBD (without EST). Since Jeong reported on the safety and efficacy of EPLBD (without EST) for the management of large CBDS, more recent studies have confirmed the safety and effectiveness of EPLBD for removing large CBDS [12,13]. Currently, both procedures are available for large CBDS, but in many institutions, EPLBD is indicated only in patients with a bleeding tendency.

As described above, there are various methods for removing stones and many comparative studies have been performed. Several meta-analyses have been reported [14,15,16], but most were pairwise meta-analyses comparing EST vs. EPBD and EST vs. ESLBD. These pairwise meta-analyses are important, but it remains unclear which procedure is optimal for the management of CBDS via the papilla. We searched for published studies on this topic in electronic databases including PubMed and reviewed the studies to compare the efficacy and safety of EST vs. EPBD vs. ESBD for the removal of small CBDS and EST vs. EPLBD vs. ESLBD for large CBDS.

### 1.1. Definitions of Endoscopic Techniques

The stone removal techniques were classified into five categories: EST, EPBD, ESBD, EPLBD, and ESLBD. Each treatment was defined as follows: (1) EST: a full EST was defined as extending to almost the total length of the ampullary-protruding portion. An EST that extended over the covering fold was defined as a medium EST, and an EST that did not extend over the covering fold was defined as a small EST. (2) EPBD: EPBD using a ≤10 mm balloon. (3) ESBD: a small EST with a shorter incision of ≤4 mm was defined as minimal EST. Minimal EST followed by EPBD was defined as ESBD. (4) EPLBD: EP large BD using a ≥12 mm balloon without EST. (5) ESLBD: EPLBD with EST. Based on this classification, small stones were treated using EST, EPBD, and ESBD, and large stones using EST, EPLBD, and ESLBD. Table 1 shows the types of endoscopic procedures via the papilla based on our definitions.

### 1.2. Study Selection and Data Extraction

Both prospective randomized controlled trials (RCTs) and retrospective comparative studies were included from the search of PubMed Central with the basic keywords “common bile duct stones” and “endoscopic treatment”, and the additional keywords “sphincterotomy”, “EST”. “EPBD”, “balloon dilation,” and “EPLBD”. The literature search included papers published between January 1990 and May 2020. Selected studies compared EST vs. EPBD or EST vs. ESBD for small CBDS, and EST and ESLBD, EPLBD and ESLBD, or EST vs. EPLBD for removing large stones. If an adequate number of RCTs had been published for all procedures, we would have analyzed only RCTs. However, this was not the case, and therefore, we included retrospective comparative studies.

The diameters of the CBDS and sizes of the balloons differed among the reports, and reports in which the balloon size used was unclear were excluded [17]. We defined EPBD as using a balloon size of ≤10 mm and EPLBD as using a balloon size of ≥12 mm. Some articles compared 10 mm balloons with large balloons ≥12 mm, and these articles were excluded from this analysis [18,19,20,21,22].

We extracted the following information: baseline trial data including first author, publication year, article type, number of patients, type of endoscopic technique, balloon size, duration of BD, number of stones, diameter of stones, first/overall complete stone removal rate, requirement for ML, and AEs including bleeding, perforation, PEP, and cholangitis/cholecystitis. For the analysis of efficacy, we used the complete stone removal rate at the initial session and the rate of ML. For the analysis of the safety of the procedure, all AEs, bleeding, PEP, and perforations were included. Ease of the procedure was difficult to describe and analyze; therefore, we determined the ease of procedure based on our experience. There were two categories: ease of the procedure in terms of papillary access and ease of stone removal.

### 1.3. Statistical Analyses

Statistical analyses were performed using RevMan software (Review Manager Version 5.4, Nordic Cochrane Centre, Copenhagen, Denmark). For comparing dichotomous variables in each trial, odds ratios (ORs) and 95% confidence intervals (95%CIs) were calculated. Statistical heterogeneity among trials was evaluated by Cochrane Q with the chi-squared test, and statistical significance was defined as *p* < 0.05.

## 2. Results

### 2.1. Removal of Small CBDS

Table 2 shows the therapeutic results and complications from treatment with EST, EPBD, and ESBD for removing small stones. There were 11 RCTs [5,7,23,24,25,26,27,28,29,30,31] comparing EST with EPBD, and one retrospective non-randomized comparative study that compared EST with ESBD.

### 2.2. EST vs. EPBD

We found 11 RCTs comparing EST (*n* = 779) with EPBD (*n* = 766) for the removal of small CBDS, with no retrospective study in this analysis. Figure 1 shows a forest plot comparing EST and EPBD in terms of complete stone removal rate at the first session, rate of ML, and rates of overall AEs, bleeding, PEP, and perforations.

**Efficacy:** One of the 11 RCTs reported that the overall stone removal rate was significantly higher with EST than with EPBD [30], but no significant differences were reported in the other 10 RCTs. Both EST and EPBD resulted in relatively good overall complete stone removal rates: 86.9–100% for EST, and 86.7–100% for EPBD. Two of the 11 RCTs reported that the complete stone removal rate at the first session was significantly higher for EST [24,25]. From the analysis of pooled data from 11 RCTs, complete stone removal rate at the first session was superior for EST compared to EPBD (OR = 0.77, 95%CI: 0.60–0.99, *p* = 0.04, I^2^ = 61%). In three of the 11 RCTs, the rate of ML was significantly lower for EST than for EPBD [23,26,30]. EPBD was associated with more frequent ML when removing CBDS (OR = 1.82, 95%CI: 1.36–2.42, *p* <0.001, I^2^ = 31%) in the pooled analysis.

**Safety:** Regarding AEs, in two RCTs, the overall rate of AEs was higher for EPBD than for EST [5,30]. Of note, four RCTs revealed that PEP incidence was significantly higher for EPBD [5,7,25,30]. There, 11 RCTs reported no bleeding from EPBD, significantly lower than the incidence with EST. The rate of perforation was extremely low for both EST and EPBD. EPBD is thought to not cause perforation, but one study did report that perforation occurred with EPBD.

Pooled analysis showed EPBD was associated with a higher incidence of overall AEs (OR = 1.46, 95%CI: 1.05–2.04, *p* = 0.03, I^2^ = 55%) than EST was, particularly PEP (OR = 2.53, 95%CI: 1.64–3.91, *p* <0.001, I^2^ = 39%). Bleeding was more frequent in EST than in EPBD (OR = 0.20, 95%CI: 0.06–0.63, *p* = 0.006, I^2^ = 0%). There was no significant difference in the rate of perforation between the two techniques (OR = 0.65, 95%CI: 0.17–2.52, *p* = 0.73, I^2^ = 0%).

### 2.3. EST vs. ESBD

Only one non-RCT (NRCT) [8] was reported from our institution (Table 2).

**Efficacy:** There were no significant differences in the overall complete stone removal rate, but complete stone removal rate at the first session was significantly higher for ESBD (EST 64.9% vs. ESBD 87.2%, *p* < 0.01). In addition, the rate of ML was significantly lower for ESBD (EST 16.7% vs. ESBD 7.8%, *p* = 0.007).

**Safety:** The frequency of overall AEs was significantly higher for EST than for ESBD (EST 15.8% vs. ESBD 4.4%, *p* < 0.001). There were no significant differences in the frequency of PEP, perforation, or cholangitis. Bleeding was significantly more frequent in EST than in ESBD (EST 9.6% vs. ESBD 1.2%, *p* < 0.001).

### 2.4. Removal of Large CBDS

Table 3 shows the therapeutic results and complications with EST, EPLBD, and ESLBD for removing large stones. We found five RCTs [10,11,32,33,34] and five NRCTs [35,36,37,38,39] that compared EST with ESLBD, one NRCT^43^ that compared EST with EPLBD, and two RCTs [40,41] and one NRCT [42] that compared ESLBD with EPLBD.

### 2.5. EST vs. ESLBD

We found five RCTs and five NRCTs that compared EST (*n* = 680) with ESLBD (*n* = 679) for the removal of large CBDS. Figure 2 shows a forest plot comparing complete stone removal rate at the first session, rate of ML, and rates of overall AEs, bleeding, PEP, and perforation between EST and ESLBD.

**Efficacy:** One of the five NRCTs reported that the overall complete stone removal rate was significantly higher for ESLBD than for EST [38], but the rate did not differ significantly in the remaining five RCTs and four NRCTs. Both EST and ESLBD were associated with relatively good overall complete stone removal rates (EST 69.8–100%, EPBD 95.2–100%). Two of the five RCTs [10,11] and three of the five NRCTs [36,38,39] reported that complete stone removal rate at the first session was significantly higher for ESLBD than for EST. Three of the five RCTs and three of the five NRCTs reported that the rate of ML was significantly lower for ESLBD than for EST [10,11,33,35,38,39].

Based on pooled data from the five RCTs and five NRCTs, the analysis showed that complete stone removal rate at the first session was superior for ESLBD compared to EST (OR = 0.38, 95%CI: 0.27–0.53, *p* < 0.01, I^2^ = 57%). Relative to ESLBD, EST was associated with more frequent ML during CBDS removal (OR = 2.39, 95%CI: 1.76–3.24, *p* < 0.001, I^2^ = 62%).

**Safety:** Regarding AEs, one of the five RCTs reported that the rate of overall AEs was higher in EST than in ESLBD [33]. In particular, one RCT revealed that cholangitis was significantly more frequent in EST than in ESLBD [33]. The rates of bleeding, PEP, and perforation were not significantly different between EST and ESLBD.

From our analysis, there was no significant difference in the incidence of overall AEs (OR = 1.33, 95%CI: 0.94–1.87, *p* = 0.10, I^2^ = 15%), bleeding (OR = 1.26, 95%CI: 0.75–2.10, *p* = 0.38, I^2^ = 21%), PEP (OR = 1.08, 95%CI: 0.65–1.79, *p* = 0.77, I^2^ = 0%), or perforation (OR = 1.78, 95%CI: 0.48–6.62, *p* = 0.39, I^2^ = 0%) between EST and ESLBD.

### 2.6. EST vs. EPLBD

Only one RCT comparing EST and EPLBD was reported by Kogure [43].

**Efficacy:** There were no significant differences in the overall complete stone removal rate (EST 95.3% vs. EPLBD 100%, *p* = 0.06), but the complete stone removal rate at the first session was significantly higher (EST 78.8% vs. EPLBD 90.7%, *p* = 0.04), and the rate of ML was significantly lower (EST 69.2% vs. EPLBD 43.9%, *p* = 0.02), in EPLBD.

**Safety:** There was no significant difference in the frequency of overall AEs (EST 9.4% vs. EPLBD 9.3%), bleeding (1.2% vs. 0%), PEP (5.9% vs. 4.7%), perforation (0% vs. 0%), or cholangitis (2.4% vs. 3.5%).

### 2.7. EPLBD vs. ESLBD

We found two RCTs and one NRCT comparing EPLBD (*n* = 132) with ESLBD (*n* = 127) for the removal of large CBDS. Figure 3 shows a forest plot comparing complete stone removal rate at the first session, the rate of ML, and rates of overall AEs, bleeding, PEP, and perforation between EPLBD with ESLBD.

**Efficacy:** The overall complete stone removal rate (EPLBD 96.4–97.6% vs. ESLBD 95.7–100%), complete stone removal rate at the first session (89.3–95.2% vs. 71.4–97.7%), and rate of ML (10.7–21.4% vs. 7.1–26.1%) did not differ significantly between EPLBD and ESLBD in the two RCTs and one NRCT [40,41,42]. From the pooled data of the three studies, there was no significant difference in the complete stone removal rate at the first session (OR = 1.09, 95%CI: 0.43–2.78, *p* = 0.85, I^2^ = 23%) or the rate of ML (OR = 0.98, 95%CI: 0.52–1.84, *p* = 0.95, I^2^ = 0%).

**Safety:** The rates of overall AEs, bleeding, perforation, PEP, and cholangitis did not differ significantly between EPLBD and ESLBD. From our analysis, there was no significant difference in the incidence of overall AEs (OR = 1.07, 95%CI: 0.43–2.65, *p* = 0.88, I^2^ = 0%), bleeding (OR = 1.05, 95%CI: 0.06–17.33, *p* = 0.97), PEP (OR = 0.87, 95%CI: 0.32–2.33, *p* = 0.78, I^2^ = 0%), or perforation (OR = 0.74, 95%CI: 0.09–5.97, *p* = 0.78, I^2^ = 0%).

## 3. Discussion

Endoscopic management of CBDS is a standard and widely accepted procedure. There are various procedures via the papilla to open the orifice of the bile duct for the removal of CBDS. We considered the efficacy of stone removal, safety, and ease of the procedure, to select the best procedure according to an optimal balance in these items. We reviewed comparative studies of these procedures for both small and large CBDS. From our review, ESBD for the management of small CBDS and ESLBD for large CBDS are considered the best procedures, combining small sphincterotomy and BD. A small incision is considered best for the prevention of PEP and a relatively easy procedure. Bleeding and perforation were rare with smaller incisions. Additional BD may make the stone removal procedure easier than with simple EST.

### 3.1. Small CBDS

Several RCTs comparing EST and EPBD have been reported. Comparing EST and EPBD in our analysis, the results (Figure 1) indicate that EST is associated with a higher complete stone removal rate at the first session, less frequent ML while removing CBDS, and a lower incidence of overall AEs than EPBD. Regarding AEs, the incidence of pancreatitis was significantly higher in EPBD, and bleeding was more frequent in EST. Therefore, EST is considered a more efficient and safer technique than EPBD, excluding the risk of bleeding.

ESBD was developed to overcome the risk of bleeding without increasing the risk of PEP, because the pancreatic orifice is separated from the biliary orifice after minimal EST. ESBD is easy to perform for beginners because it requires only a minimal incision. In addition, ESBD is expected to result in a wider opening, facilitating stone removal [44]. ESBD has not been reported in any RCT, and there is only one NRCT on this topic from our institution [8]. From our data, complete stone removal rate at the first session was significantly higher, the rate of ML was lower, and the incidence of overall AEs, particularly bleeding, was lower in ESBD compared with EST. ESBD was more efficient and safe in the management of CBDS than EST was. The number of patients taking antithrombotic drugs is increasing as the elderly population increases, and most patients cannot stop these drugs safely due to the high risk of cardiovascular or thromboembolic events [45,46]. Management of the risk of bleeding will be increasingly important in future. ESBD may help with overcoming the risk of bleeding, which is a disadvantage of EST. There are no RCTs comparing EST with ESBD, and the evidence is limited. We plan to conduct a prospective multicenter RCT to evaluate this combined technique.

### 3.2. Large CBDS

There are three methods for removing large stones: EST, EPLBD, and ESLBD. Since Ersoz first reported the use of ESLBD in patients in whom EST and EPBD had failed to remove CBDS [9], ESLBD has become widespread, and RCTs comparing ESLBD with EST have been subsequently reported. In the comparison of EST and ESLBD (Figure 2), ESLBD was associated with a higher complete stone removal rate at the first session and less frequent ML use. There was no significant difference in the incidence of overall AEs, bleeding, PEP, or perforation. Therefore, ESLBD is considered a more efficient and equally safe technique compared with EST. In contrast to ESLBD, which requires a small incision, Jeong reported on the safety and efficacy of EPLBD (without a preceding EST) in 2009 [13], suggesting that a preceding EST may not play a significant role in the guidance of BD. Based on that report, one RCT compared EST with EPLBD. In the comparison of EST and EPLBD, there was no significant difference in the overall complete stone removal rate, but complete stone removal rate at the first session was significantly higher in EPLBD, and the rate of ML was significantly lower, without increasing the risk of PEP. From comparing EST vs. ESLBD and EST vs. EPLBD, both ESLBD and EPLBD are considered more efficient and safer than EST. However, which technique is better for removing large CBDS, EPLBD or ESLBD? Two RCTs and one NRCT comparing EPLBD and ESLBD have been published. In our analysis, there was no significant difference between EPLBD and ESLBD in terms of complete stone removal rate at the first session, rate of ML, and incidence of overall AEs, bleeding, PEP, or perforation. In ESLBD, the additional EST may separate the pancreatic and biliary orifices and the BD forces are further away from the pancreatic duct, resulting in a decreased risk of PEP. EPBD is associated with high risk for PEP, and the small BD cannot separate the pancreatic orifice from the biliary orifice. However, accumulating EPLBD data have not shown an increased risk of PEP. EPBD and EPLBD differ clinically. An insufficiently enlarged bile duct orifice hampers the insertion of endoscopic devices and subsequent stone removal, and this stress on the pancreatic orifice results in PEP. EPLBD can dilate the orifice adequately, and the pancreatic orifice may be separated from the biliary orifice due to the large BD. Based on our study, EPLBD may be superior to ESLBD in terms of being simpler and easier to perform, provided that the efficiency of stone removal and incidence of AEs including PEP are comparable to those of ESLBD. However, there are only two RCTs and one NRCT with a relatively small number (42–69) of cases comparing EPLBD and ESLBD; therefore, additional RCTs with larger sample sizes are needed to evaluate the safety and efficacy of EPLBD. Although several RCTs comparing EST vs. ESLBD, EPLBD vs. ESLBD, and EST vs. EPLBD have been published, there are no comprehensive results comparing the three endoscopic techniques. Guo reported on an RCT that compared EST with EPLBD and ESLBD, but the balloon was 10–15 mm in diameter, and this RCT was excluded from our study [20]. There was only one NRCT by Kuo et al. comparing EST, EPLBD, and ESLBD for large CBDS that met our criteria [47]. The authors reported that ESLBD led to similar outcomes in terms of overall complete stone removal rate and the rate of ML, but it had a higher complete stone removal rate at the first session compared with EST and EPLBD. Regarding AEs, there was no significant difference in the incidence of PEP, perforation, or cholangitis among the three groups, but the incidence of bleeding was higher in EST than in ESLBD. Based on these results, ESLBD is considered the best procedure for the management of large CBDS. EPLBD can be performed in cases in which EST would be difficult due to coagulopathy or when sphincterotome control is difficult.

Table 4 shows the terms of comparison for each procedure; i.e., ease of the procedure, efficacy, and safety in removing small and large CBDS. Table 1 compared EST with EPBD, EST with ESBD, and Table 2 compared EST with ESLBD, EPLBD with ESLBD, and EST with EPLBD. The results of complete stone removal in initial session, rate of ML related to efficiency and adverse events rates related to safety are summarized in Table 4 for each technique in each table. Table 5 is based on the results of Table 4, and scores of 1~3 are assigned according to the size of the CBDS. The ease of the procedure and stone extraction was difficult to evaluate and analyze; therefore, we determined ease based on our experience. Comparing the incision length for removal of small CBDS, ESBD requires minimal EST, which is easier than conventional EST. Stone extraction after ESBD is easier than in EST because of the larger opening of the orifice of the papilla. EPBD is a simple procedure and easier to perform than EST. However, stone extraction after EPBD is more difficult than after EST because of the small opening of the orifice of the papilla. To remove large CBDS, the adequate incision length for EST is relatively longer, making the technique more difficult, and causing an increase in AEs. By contrast, a small incision is recommended for ESLBD to avoid bleeding and perforation. It is technically easier to remove large CBDS with ESLBD, because after the procedure with the papilla, the opening is larger with ESLBD, making stone removal easier than after EST. Comparing EST and EPLBD, EST requires a relatively long incision for the removal of large CBDS, and EPLBD is considered a simpler and easier procedure. Stone removal after EPLBD is easier than it is after EST because the opening of the papilla is larger with EPLBD. Comparing ESLBD and EPLBD, ESLBD requires sphincterotomy, making EPLBD a simpler and easier procedure. From our experience, the technique of stone extraction with ESLBD is similar to EPLBD, but sometimes easier. The additional EST may play an important role regarding the papillary opening. According to Table 4, the total scores of EST, EPBD, and ESBD were 13, 12, and 18, respectively, with ESBD scoring the highest for the removal of small CBDS. For large CBDS, EST, EPLBD, and ESLBD scored 9, 17, and 17 points, respectively, with EPLBD and ESLBD scoring the highest. ESLBD and EPLBD cases were reported for a total of 806 and 132 patients, respectively; thus, there is more evidence available for ESLBD, which currently has been reported for more patients. For the management of small and large CBDS, ESBD and ESLBD are considered the best procedures, respectively.

Recently, devices integrating a sphincterotome and a balloon (StoneMaster V; Olympus Corp., Tokyo, Japan) have been developed, making it easier to perform ESBD and ESLBD (Figure 4). With the development of this device, not only ESLBD but also ESBD will become popular.

## 4. Limitations

This study had several limitations. First, various biases cannot be excluded because the study included both RCTs and NRCTs. Sources of heterogeneity that may have caused bias included (1) variation in the length of EST among studies, (2) different durations of BD, (3) the absence of difficult cannulation cases such as periampullary diverticulum in some reports, and (4) differences in indications for ML and assessment of minor bleeding among individual endoscopists. Second, the number of cases comparing EPLBD vs. ESLBD and EST vs. ESBD included in this study was small. In particular, ESBD was only reported in one NRCT from our institution. Third, we defined EPBD and EPLBD as procedures using balloon sizes ≤10 mm and ≥12 mm, respectively; however, some papers defined large balloons as >10 mm [20,21]. This difference in definition might have led to selection bias. Finally, reports in languages other than English were excluded.

## 5. Conclusions

ESBD is considered the best procedure for the management of small CBDS. An RCT comparing ESBD with EST is warranted to confirm this result. For large CBDS, ESLBD is considered the best procedure. Further RCTs comparing EPLBD with ESLBD are required to evaluate the safety of EPLBD.

## Figures and Tables

**Figure 1 jcm-09-03808-f001:**
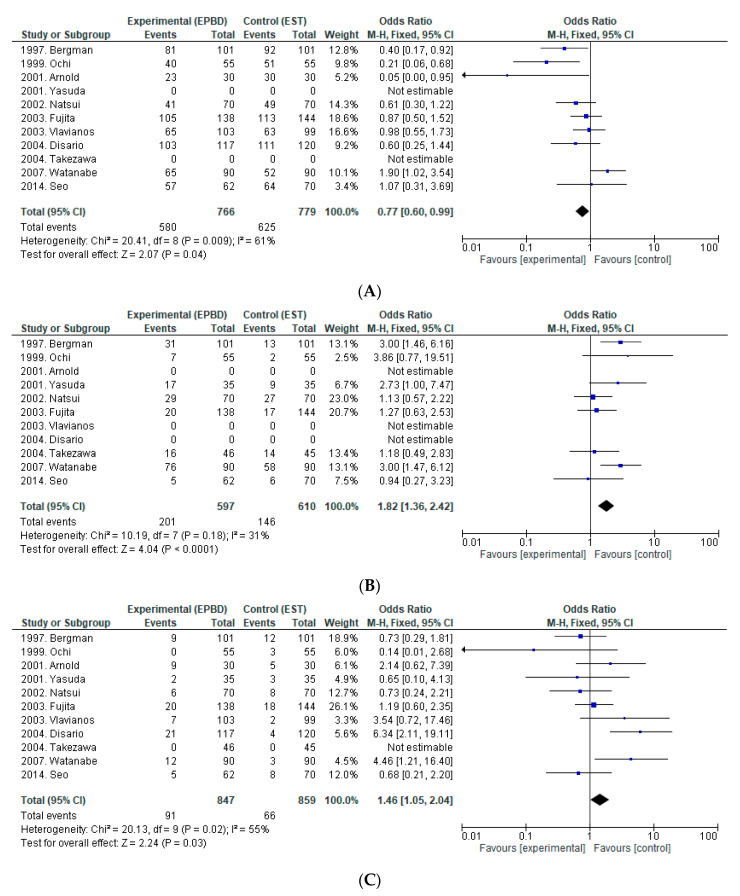
Forest plot of risk ratios and 95%CIs representing a comparison between EST and EPBD. (**A**): Complete stone removal at the first endoscopic session (**B**): Rate of mechanical lithotripsy. (**C**): Overall adverse events. (**D**): Bleeding. (**E**): PEP. (**F**): Perforation. CI, Confidence interval; EST, endoscopic sphincterotomy; EPBD, endoscopic papillary balloon dilation; PEP, post-endoscopic retrograde cholangiopancreatography pancreatitis.

**Figure 2 jcm-09-03808-f002:**
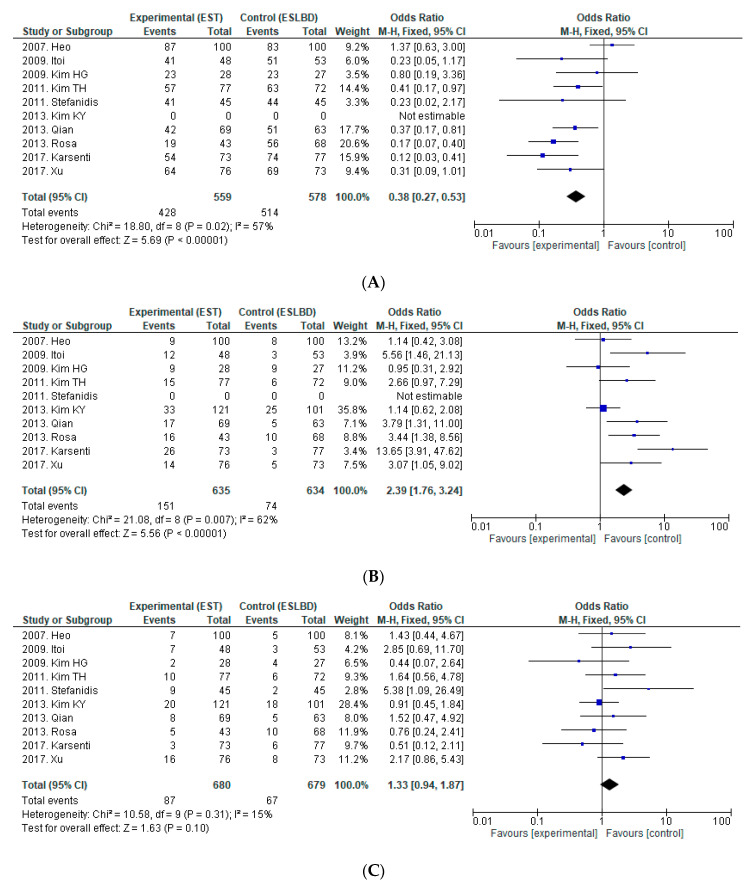
Forest plot of risk ratios and 95%CIs representing a comparison between EST and ESLBD. (**A**): Complete stone removal in the first endoscopic session. (**B**): Rate of mechanical lithotripsy. (**C**): Overall adverse events. (**D**): Bleeding. (**E**): PEP. (**F**): Perforation. CI, Confidence interval; EST, endoscopic sphincterotomy; ESLBD, endoscopic papillary large balloon dilation with endoscopic sphincterotomy; PEP, post-endoscopic retrograde cholangiopancreatography pancreatitis.

**Figure 3 jcm-09-03808-f003:**
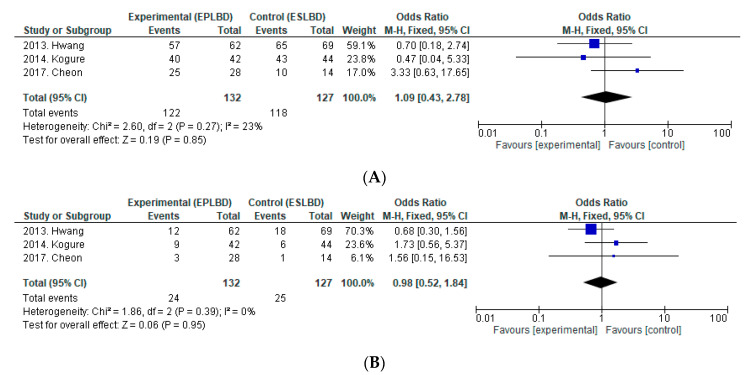
Forest plot of risk ratios and 95%CIs representing a comparison between EPLBD and ESLBD. (**A**): Complete stone removal in the first endoscopic session. (**B**): Rate of mechanical lithotripsy. (**C**): Overall adverse events. (**D**): Bleeding. (**E**): PEP. (**F**): Perforation. CI, Confidence interval; EPLBD, endoscopic papillary large balloon dilation without endoscopic sphincterotomy; ESLBD, endoscopic papillary large balloon dilation with endoscopic sphincterotomy; PEP, post-endoscopic retrograde cholangiopancreatography pancreatitis.

**Figure 4 jcm-09-03808-f004:**
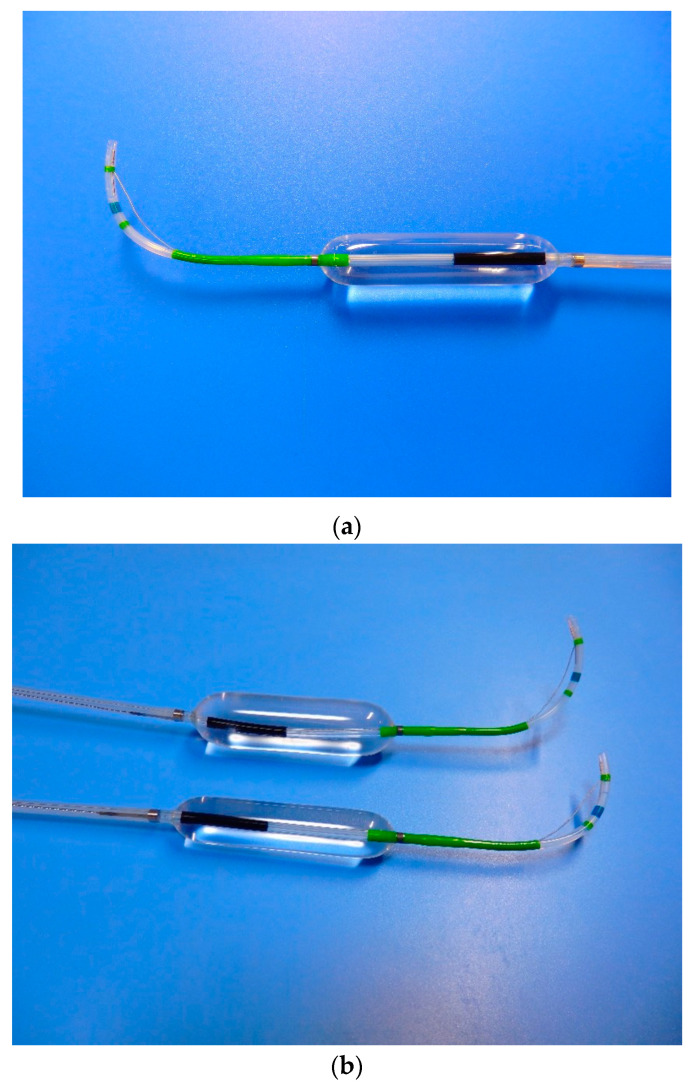
(**a**) A device integrating a sphincterotome and a balloon (StoneMaster V; Olympus Corp., Tokyo, Japan). (**b**) Two types of balloon sizes that can be used for ESBD and ESLBD. ESBD, endoscopic papillary balloon dilation with endoscopic sphincterotomy; ESLBD, endoscopic papillary large balloon dilation with endoscopic sphincterotomy.

**Table 1 jcm-09-03808-t001:** Types of endoscopic procedure on the papilla.

		Balloon Dilation		
		(-)	≤10 mm	12 mm≤
Sphincterotomy	(+)	EST	ESBD	ESLBD
	(-)	-	EPBD	EPLBD

EST; endoscopic sphincterotomy, EPBD; endoscopic papillary balloon dilation, ESBD; EST followed by balloon dilation, EPLBD; endoscopic papillary large balloon dilation, ESLBD; EST followed by large balloon dilation.

**Table 2 jcm-09-03808-t002:** Baseline characteristics, procedural outcomes and adverse events of the included studies for small common bile dust stones.

Author	Comparison	Study Design	No. of Patients	Diameter of Stones (mm) Mean ± SD or Mean (Range)	Balloon Size, mm	Procedure Time, min	ML Use % (No.)	Successful Stone Removal in the 1st Session % (No.)	Overall Uccessful Stone Removal % (No.)	Overall AEs % (No.)	Bleeding % (No.)	Perforation % (No.)	PEP % (No.)	Cholangitis % (No.)
**EST vs. EPBD**														
Bergman	EST	RCT	101	9 (4–27)		NA	12.9% (13)	91.1% (92)	91.1% (92)	11.9% (12)	4% (4)	1% (1)	6.9% (7)	0% (0)
	EPBD	RCT	101	10 (3–36)	8	NA	30.7% (31)	80.2% (81)	89.1% (90)	8.9% (9)	0% (0)	2% (2)	6.9% (7)	0% (0)
Ochi	EST	RCT	55	8.8 ± 4.2		NA	3.7% (2)	94.4% (51)	98.1% (54)	5.6% (3)	0% (0)	1.8% (1)	3.6% (2)	0% (0)
	EPBD	RCT	55	8.1 ± 3.4	8	NA	13.7% (7)	78.4% (40)	92.7% (51)	0% (0)	0% (0)	0% (0)	0% (0)	0% (0)
Arnold	EST	RCT	30	10 ± 4.7		17 ± 12	NA	100% (30)	100% (30)	16.7% (5)	6.7% (2)	0% (0)	10% (3)	0% (0)
	EPBD	RCT	30	7 ± 3.5	8	29 ± 15	NA	76.7% (23)	100% (30)	30% (9)	0% (0)	0% (0)	20% (6)	10% (3)
Yasuda	EST	RCT	35	12.3 (5–24)		NA	25.7% (9)	100% (35)	100% (35)	8.6% (3)	2.9% (1)	0% (0)	5.7% (2)	0% (0)
	EPBD	RCT	35	12.4 (4–24)	8	NA	48.6% (17)	100% (35)	100% (35)	5.7% (2)	0% (0)	0% (0)	5.7% (2)	0% (0)
Natsui	EST	RCT	70	9.7 (3–17)		NA	38.6% (27)	70% (49)	98.6% (69)	11.4% (8)	2.9% (2)	0% (0)	4.3% (3)	4.3% (3)
	EPBD	RCT	70	9.2 (3–22)	8	NA	41.4% (29)	58.6% (41)	92.9% (65)	8.6% (6)	0% (0)	0% (0)	5.7% (4)	2.9% (2)
Fujita	EST	RCT	144	7.3 ± 3.4		NA	11.8% (17)	78.5% (113)	100% (144)	12.5% (18)	1.4% (2)	0% (0)	2.8% (4)	8.3% (12)
	EPBD	RCT	138	7 ± 3.1	4-8	NA	14.5% (20)	76.1% (105)	99.3% (137)	14.5% (20)	0% (0)	0% (0)	10.9% (15)	3.6% (5)
Vlavianos	EST	RCT	99	NA		NA	NA	63.6% (63)	86.9% (86)	2% (2)	0% (0)	0% (0)	1% (1)	1% (1)
	EPBD	RCT	103	NA	10	NA	NA	63.1% (65)	87.4% (90)	6.8% (7)	0% (0)	0% (0)	4.9% (5)	1.9% (2)
Disario	EST	RCT	120	5 (0.5–14)		42 ± 19	NA	92.5% (111)	NA	3.3% (4)	0% (0)	0.8% (1)	0.8% (1)	1.7% (2)
	EPBD	RCT	117	6 (0.5–10)	≤8	47 ± 25	NA	88% (103)	NA	17.9% (21)	0% (0)	0% (0)	15.4% (18)	2.6% (3)
Takezawa	EST	RCT	45	11 (3–27)		37.2 ± 12.4	31.1% (14)	NA	100% (45)	0% (0)	0% (0)	0% (0)	0% (0)	0% (0)
	EPBD	RCT	46	10 (1–35)	8	37.4 ± 13	34.8% (16)	NA	100% (46)	0% (0)	0% (0)	0% (0)	0% (0)	0% (0)
Watanabe	EST	RCT	90	7.7 ± 2.9		NA	64.4% (58)	57.8% (52)	95.6% (86)	3.3% (3)	1.1% (1)	0% (0)	2.2% (2)	0% (0)
	EPBD	RCT	90	8.1 ± 3.2	8	NA	84.4% (76)	72.2% (65)	86.7% (78)	13.3% (12)	0% (0)	0% (0)	10% (9)	3.3% (3)
Seo	EST	RCT	70	7.6 ± 3.1		NA	8.6% (6)	91.4% (64)	100% (70)	11.4% (8)	2.9% (2)	1.4% (1)	7.1% (5)	0% (0)
	EPBD	RCT	62	7.2 ± 2.1	6-10	NA	8.1% (5)	91.9% (57)	98.4% (61)	8.1% (5)	0% (0)	0% (0)	8.1% (5)	0% (0)
**EST vs. ESBD**														
Ishii	EST	NRCT	114	5.1 ± 2.9		31.6 ± 16.8	16.7% (19)	64.9% (74)	100% (114)	15.8% (18)	9.6% (11)	1.8% (2)	2.6% (3)	1.8% (2)
	ESBD	NRCT	321	5.6 ± 3	8-10	25.8 ± 18.8	7.8% (25)	87.2% (280)	100% (321)	4.4% (14)	1.2% (4)	0.3% (1)	1.9% (6)	0.9% (3)

**Table 3 jcm-09-03808-t003:** Baseline characteristics, procedural outcomes and adverse events of the included studies for large common bile dust stones.

Author	Comparison	Study Design	No. of Patients	Diameter of Stones (mm) mean ± SD or mean (Range)	Balloon Size, mm	Procedure Time, min	ML Use % (No.)	Successful Stone Removal in the 1st Session % (No.)	Overall Uccessful Stone Removal % (No.)	Overall AEs % (No.)	Bleeding % (No.)	Perforation % (No.)	PEP % (No.)	Cholangitis % (No.)
**EST vs. ESLBD**														
Heo	EST	RCT	100	15 ± 0.7		NA	9% (9)	87% (87)	98% (98)	7% (7)	2% (2)	0% (0)	4% (4)	1% (1)
	ESLBD	RCT	100	16 ± 0.7	12–20	NA	8% (8)	83% (83)	97% (97)	5% (5)	0% (0)	0% (0)	4% (4)	1% (1)
Kim	EST	RCT	28	21.3 ± 5.2		19 ± 13	32% (9)	82.1% (23)	100% (28)	7.1% (2)	7.1% (2)	0% (0)	0% (0)	0% (0)
	ESLBD	RCT	27	20.8 ± 4.1	15–18	18 ± 12	33% (9)	85.2% (23)	100% (27)	14.8% (4)	14.8% (4)	0% (0)	0% (0)	0% (0)
Stefanidis	EST	RCT	45	NA		NA	100% (45)	91.1% (41)	91.1% (41)	20% (9)	2.2% (1)	2.2% (1)	2.2% (1)	13.3% (6)
	ESLBD	RCT	45	NA	15–20	NA	0% (0)	97.7% (44)	97.8% (44)	4.4% (2)	2.2% (1)	0% (0)	2.2% (1)	0% (0)
Qian	EST	RCT	69	20.3 ± 5.3		15.9 ± 8.8	24.6% (17)	60.9% (42)	91.3% (63)	11.6% (8)	0% (0)	1.4% (1)	8.7% (6)	1.4% (1)
	ESLBD	RCT	63	20.6 ± 5.4	12–20	14.5 ± 8.4	7.9% (5)	81% (51)	95.2% (60)	7.9% (5)	0% (0)	0% (0)	6.3% (4)	1.6% (1)
Karsenti	EST	RCT	73	16.2 ± 3.5		NA	35.6% (26)	74% (54)	94.5% (69)	4.1% (3)	2.7% (2)	0% (0)	0% (0)	1.4% (1)
	ESLBD	RCT	77	16.5 ± 3.3	12–20	NA	3.9% (3)	96.1% (74)	96.1% (74)	7.8% (6)	3.9% (3)	1.3% (1)	1.3% (1)	1.3% (1)
Itoi	EST	NRCT	48	15.3 ± 3.2		40.2 ± 16.3	25% (12)	85.4% (41)	97.9% (47)	14.6% (7)	8.3% (4)	0% (0)	4.1% (2)	2.1% (1)
	ESLBD	NRCT	53	14.8 ± 3.5	15–20	31.6 ± 11.3	5.7% (3)	96.2% (51)	100% (53)	5.7% (3)	1.9% (1)	0% (0)	1.9% (1)	1.9% (1)
Kim KY	EST	NRCT	121	10 (2–20)		NA	27.3% (33)	NA	100% (121)	16.5% (20)	14% (17)	0.8% (1)	1.7% (2)	0% (0)
	ESLBD	NRCT	101	12 (3–25)	12–	NA	24.8% (25)	NA	99% (100)	17.8% (18)	16.8% (17)	0% (0)	1% (1)	0% (0)
Kim TH	EST	NRCT	77	NA		NA	19.5% (15)	74% (57)	94.8% (73)	13% (10)	0% (0)	1.3% (1)	11.7% (9)	0% (0)
	ESLBD	NRCT	72	NA	12–20	NA	8.3% (6)	87.5% (63)	97.2% (70)	8.3% (6)	0% (0)	0% (0)	6.9% (5)	1.3% (1)
Xu	EST	NRCT	76	16.5 ± 4.7		47.3 ± 11.8	18.4% (14)	84.2% (64)	100% (76)	21.1% (16)	7.9% (6)	0% (0)	9.2% (7)	4% (3)
	ESLBD	NRCT	73	16.9 ± 4.1	12–20	42.1 ± 13.6	6.8% (5)	94.5% (69)	100% (73)	11% (8)	1.4% (1)	0% (0)	8.2% (6)	1.4% (1)
Rosa	EST	NRCT	43	16 ± 6.7		NA	37.2% (16)	44.2% (19)	69.8% (30)	11.6% (5)	4.7% (2)	0% (0)	4.7% (2)	2.3% (1)
	ESLBD	NRCT	68	16.8 ± 4.4	12-18	NA	14.7% (10)	82.4% (56)	95.6% (65)	14.7% (10)	0% (0)	0% (0)	13.2% (9)	1.5% (1)
**EPLBD vs. ESLBD**														
Hwang	EPLBD	RCT	62	15.7 ± 3.3	12–20	NA	19.4% (12)	91.9% (57)	96.8% (60)	6.5% (4)	0% (0)	0% (0)	6.5% (4)	0% (0)
	ESLBD	RCT	69	16.5 ± 4.2	12–20	NA	26.1% (18)	94.2% (65)	95.7% (66)	5.7% (4)	0% (0)	1.4% (1)	4.3% (3)	0% (0)
Cheon	EPLBD	RCT	42	14.4 ± 3.3	12–	10.8 ± 6.9	21.4% (9)	95.2% (40)	97.6% (41)	11.9% (5)	2.4% (1)	0% (0)	7.1% (3)	2.4% (1)
	ESLBD	RCT	44	14 ± 2.1	12–	10.6 ± 5.7	13.6% (6)	97.7% (43)	100% (43)	11.4% (5)	2.3% (1)	0% (0)	11.4% (5)	0% (0)
Kogure	EPLBD	NRCT	28	14 ± 4	12–18	NA	10.7% (3)	89.3% (25)	96.4% (27)	7.1% (2)	0% (0)	3.6% (1)	3.6% (1)	0% (0)
	ESLBD	NRCT	14	14 ± 4	12–18	NA	7.1% (1)	71.4% (10)	100% (14)	7.1% (1)	0% (0)	0% (0)	7.1% (1)	0% (0)
**EST vs. EPLBD**														
Kogure	EST	RCT	85	14.3 ± 4.8		52.6 ± 58.4	48.2% (41)	78.8%(67)	95.3% (81)	9.4% (8)	1.2% (1)	0% (0)	5.9% (5)	2.4% (2)
	EPLBD		86	15.2 ± 4.6		49.6 ± 39.7	30.2% (26)	90.7% (78)	100% (86)	9.3% (8)	0% (0)	0% (0)	4.7% (4)	3.5% (3)

**Table 4 jcm-09-03808-t004:** Comparison of the procedures on the papilla. Rate of each procedure based on the results of the analysis (Figure 1, Figure 2 and Figure 3).

	Removal of Small Stones			Removal of Large Stone		
	EST	EPBD	ESBD	EST	EPLBD	ESLBD
Efficacy						
Complete stone removal in initial session	78.3% (699/893)	75.7% (580/766)	87.2 (280/321)	76.6% (428/559)	92.4% (122/132)	89.6% (632/705)
Rate of ML	22.8% (165/724)	33.7% (201/597)	7.8% (25/321)	23.8% (151/635)	18.2% (24/132)	13.0% (99/761)
Safety						
Bleeding	2.6% (25/973)	0% (0/847)	1.2% (4/321)	5.3% (36/680)	0.8% (1/132)	3.5% (28/806)
Post ERCP pancreatitis	4.9% (48/973)	8.4% (71/847)	4.4% (14/321)	4.9% (33/680)	6.1% (8/132)	5.1% (41/806)
Perforation	0.6% (6/973)	0.2% (2/847)	0.3% (1/321)	0.6% (4/680)	0.8% (1/132)	0.2% (2/806)

EST; endoscopic sphincterotomy, EPBD; endoscopic papillary balloon dilation, ESBD; EST followed by balloon dilation, EPLBD; endoscopic papillary large balloon dilation, ESLBD; EST followed by large balloon dilation.

**Table 5 jcm-09-03808-t005:** Ranking in efficacy, safety and ease of the procedure.

	Removal of Small Stones			Removal of Large Stone		
	EST	EPBD	ESBD	EST	EPLBD	ESLBD
Efficacy						
Complete stone removal in initial session	2	1	3	1	3	3
Rate of ML	2	1	3	1	2	3
Safety						
Bleeding	1	3	2	1	3	2
Post ERCP pancreatitis	3	1	3	2	2	2
Perforation	2	2	2	2	2	2
Easiness						
Easiness of the procedure	1	3	2	1	3	2
Easiness of stone extraction	2	1	3	1	2	3
Total score	13	12	18	9	17	17

EST; Endoscopic Sphincterotomy, EPBD; Endoscopic papillary balloon dilation, ESBD; EST followed by balloon dilation, EPLBD; Endoscopic papillary large balloon dilation, ESLBD; EST followed by large balloon dilation. Score 3 is the best score. Scores for ease of the procedure and stone extraction were determined based on our experiences.

## Data Availability

The datasets generated and analyzed during the current study are available from the corresponding author on reasonable request.

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
