# Peer review of "Best Procedure for the Management of Common Bile Duct Stones via the Papilla: Literature Review and Analysis of Procedural Efficacy and Safety"

_jcm, 2020, doi:10.3390/jcm9123808_

Round 1

Reviewer 1 Report

This is a meta-analysis of procedural efficacy and safety aimed to reveal the best procedure for the management of common bile duct stones via the papilla.

Endoscopic procedures were classified into five categories according to endoscopic sphincterotomy (EST) and balloon dilation (BD):

1) EST

2) Endoscopic papillary BD (≤10 mm) (EPBD)

3) EST followed by BD (≤10 mm) (ESBD)

4) Endoscopic papillary large BD (≥12 mm) (EPLBD)

5) EST followed by large BD (≥12 mm) (ESLBD).

The authors performed a systematic review of 19 prospective and seven retrospective studies with a total of 3930 patients.

They concluded that ESBD is considered the best procedure for the management of small CBDS. And for large CBDS, both ESLBD and EPLBD are similar.

This analysis is very well summarized.

I think it is clinically useful, especially the fact that it is evaluated by stone size.

Please revise the following points.

Minor

1

In Table 1, both Balloon dilation (-) and Sphincterotomy (-) are marked (-), which is confusing.

Please consider changing it to blank or -.

2

Table 2 and 3 should be changed so that the tables are oriented so that they can be read all on one screen.

3

The criteria for Ranking in Table 4 are inadequately stated and the rationale should be more detailed.

4

The authors present Devices integrating a sphincterotome and a balloon (StoneMaster V; Olympus Corp., Tokyo, Japan). However, there is no data presented for this device. Considering the nature of this meta-analysis report, I think this introduction is unnecessary. Please consider this.

Author Response

Thank you for Reviewer1. The reviewers' comments were correct, and I revised the manuscript.

1

In Table 1, both Balloon dilation (-) and Sphincterotomy (-) are marked (-), which is confusing. Please consider changing it to blank or -.

→Thank you for Reviewer1. ​

I corrected the indicated part in Table 1 from (-) to -.

2

Table 2 and 3 should be changed so that the tables are oriented so that they can be read all on one screen.

→Thank you for Reviewer1.

The contents of table2 and table3 have been modified to fit everything on one screen.

3

The criteria for Ranking in Table 4 are inadequately stated and the rationale should be more detailed.

Thank you for Reviewer1.

I changed

「Table 4(a) was prepared by calculating the ratios based on the results of the meta-analysis (Figs. 1–3). Table 4(b) was based on Table 4(a), with scores of 1–3 assigned based on the size of the CBDS.」

to

「Table 1 compared EST with EPBD, EST with ESBD, and Table 2 compared EST with ESLBD, EPLBD with ESLBD, and EST with EPLBD. ​The results of complete stone removal in initial session, rate of ML related to efficiency and adverse events rates related to safety were summarized in table4(a) for each technique in each table. ​Table 4 (b) was based on the results of Table 4 (a), and scores of 1 ~ 3 are assigned according to the size of the CBDS.」

4

The authors present Devices integrating a sphincterotome and a balloon (StoneMaster V; Olympus Corp., Tokyo, Japan). However, there is no data presented for this device. Considering the nature of this meta-analysis report, I think this introduction is unnecessary. Please consider this.

→Thank you for Reviewer1. 

​ESBD for small stones is not a mainstream technique until now. ​One reason for this may be the need for multiple devices and the complexity of replacing devices.
​With the launch of this device, ESBD may become a more major technique, and we would like to present it if possible.

Reviewer 2 Report

List of corrections:

  1. Please, correct the manuscript according to PRISMA guidelines.
  2. Please add a flow diagram to depict studies included to quantitative analysis step by step.
  3. The search for studies in the area of interest should include Cochrane, ClinicalTrial.gov, Embase and Medline apart from PubMed.
  4. „Ease of the procedure” subsections do not describe results from collected data analysis but authors empirical comments an should be removed from Results section.

Author Response

Thank you for Reviewer2. The reviewers' comments were correct, and I revised the manuscript.

1. Please, correct the manuscript according to PRISMA guidelines

2. Please add a flow diagram to depict studies included to quantitative analysis step by step.

3. The search for studies in the area of interest should include Cochrane, ClinicalTrial.gov, Embase and Medline apart from PubMed.

→Thank you for Reviewer2.

Reviewer’comments 1-3 are correct and our paper does not meet the definition of meta analysis. As the reviewer pointed out, our paper is not in compliance with the PRISMA guidelines, and the studies are only searched in PubMed and quantitative analysis was not performed. We appreciate this comment and will be careful when using the term meta analysis.

​Our paper is a literature review and analysis. ​Change the subject and delete the description of the meta-analysis in the text.

Page1 Line2-4 

Best procedure for the management of common bile duct stones via the papilla: Meta-analysis of procedural efficacy and safety

→Best procedure for the management of common bile duct stones via the papilla: literature review and analysis of procedural efficacy and safety

Page2 Line45

a systematic review→a literature review 

Page19 Line272

From our meta-analysis→From our analysis

Page21 Line328

From our meta-analysis→From our analysis

Page23 Line427

Rate of each procedures based on the results of the meta-analysis

→Rate of each procedures based on the results of the analysis

4. „Ease of the procedure” subsections do not describe results from collected data analysis but authors empirical comments an should be removed from Results section.

→Thank you for Reviewer2.

We have removed all ”Ease of procedure” subsections from the Results section. We have stated on Page 23 Line 419-433 as follows.

Comparing the incision length for removal of small CBDS, ESBD requires minimal EST, which is easier than conventional EST. Stone extraction after ESBD is easier than in EST because of the larger opening of the orifice of the papilla. EPBD is a simple procedure and easier to perform than EST. However, stone extraction after EPBD is more difficult than after EST because of the small opening of the orifice of the papilla. To remove large CBDS, the adequate incision length for EST is relatively longer, making the technique more difficult, and causing an increase in AEs. By contrast, a small incision is recommended for ESLBD to avoid bleeding and perforation. It is technically easier to remove large CBDS with ESLBD, because after the procedure with the papilla, the opening is larger with ESLBD, making stone removal easier than after EST. Comparing EST and EPLBD, EST requires a relatively long incision for the removal of large CBDS, and EPLBD is considered a simpler and easier procedure. Stone removal after EPLBD is easier than after EST because the opening of the papilla is larger with EPLBD. Comparing ESLBD and EPLBD, ESLBD requires sphincterotomy, making EPLBD a simpler and easier procedure. From our experience, the technique of stone extraction with ESLBD is similar to EPLBD, but sometimes easier. The additional EST may play an important role regarding the papillary opening.